# A Non-Local Problem for the Fractional-Order Rayleigh–Stokes Equation

**Ravshan Ashurov** [1,2] 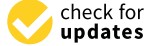, **Oqila Mukhiddinova** [1,3] and **Sabir Umarov** [4,*] 

1 Institute of Mathematics, Uzbekistan Academy of Science, University Str., 9, Olmazor District, Tashkent 100174, Uzbekistan; ashurovr@gmail.com (R.A.); oqila1992@mail.ru (O.M.)
2 AU Engineering School, Akfa University, 264, Milliy Bog Str., Tashkent 111221, Uzbekistan
3 Department of Higher Mathematics, University of Information Technologies, 108 Amir Temur Avenue, Tashkent 100200, Uzbekistan
4 Department of Mathematics, University of New Haven, 300 Boston Post Road, West Haven, CT 06516, USA
* Correspondence: sumarov@newhaven.edu; Tel.: +1-203-479-4122

**Abstract:** A nonlocal boundary value problem for the fractional version of the Rayleigh–Stokes equation, well-known in fluid dynamics, is studied. Namely, the condition $u(x,T) = \beta u(x,0) + \varphi(x)$, where $\beta$ is an arbitrary real number, is proposed instead of the initial condition. If $\beta = 0$, then we have the inverse problem in time, called the backward problem. It is well-known that the backward problem is ill-posed in the sense of Hadamard. If $\beta = 1$, then the corresponding non-local problem becomes well-posed in the sense of Hadamard, and moreover, in this case a coercive estimate for the solution can be established. The aim of this work is to find values of the parameter $\beta$, which separates two types of behavior of the semi-backward problem under consideration. We prove the following statements: if $\beta \geq 1$, or $\beta < 0$, then the problem is well-posed; if $\beta \in (0,1)$, then depending on the eigenvalues of the elliptic part of the equation, for the existence of a solution an additional condition on orthogonality of the right-hand side of the equation and the boundary function to some eigenfunctions of the corresponding elliptic operator may emerge.

**Keywords:** Rayleigh–Stokes problem; non-local problem; fractional derivative; Mittag–Leffler function; Fourier method

## 1. Introduction

Fractional derivatives serve as essential tools in the modeling of complex processes. The concept of fractional derivatives arose simultaneously with derivatives of integer order. Starting with the work of Abel (see, e.g., [1]), the concept of fractional derivatives began to be widely used in various fields, such as electrochemistry, neuron models in biology, applied mathematics, fluid dynamics, viscoelasticity and fluid mechanics [2]. Models with fractional derivatives are used to analyze the viscoelasticity, for example, of polymers during glass transition and in the glassy state [3], the theoretical base of which is the well-known Rayleigh–Stokes equation. A fractional model of a generalized second-class fluid flow can be represented as the Rayleigh–Stokes problem with a time-fractional derivative [4]:

$$\begin{cases} \partial_t u(x,t) - (1 + \gamma\,\partial_t^\alpha)\Delta u(x,t) = f(x,t), & x \in \Omega, \quad 0 < t \leq T; \\ u(x,t) = 0, & x \in \partial\Omega, \quad 0 < t \leq T; \\ u(x,0) = \varphi(x), & x \in \Omega, \end{cases} \tag{1}$$

where $1/\gamma > 0$ is the fluid density, a fixed constant; the source term $f(x,t)$ and the initial data $\varphi(x)$ are given functions, $\partial_t = \partial/\partial t$; and $\partial_t^\alpha$ is the Riemann–Liouville fractional derivative of order $\alpha \in (0,1)$ defined by (see, e.g., [1]):

$$\partial_t^\alpha h(t) = \frac{d}{dt} \int_0^t \omega_{1-\alpha}(t-s)h(s)ds, \quad \omega_\alpha(t) = \frac{t^{\alpha-1}}{\Gamma(\alpha)}. \tag{2}$$

Here $\Gamma(\cdot)$ is Euler's gamma function. Researchers, due to physical meaning, considered problem (1) in the domain $\Omega \subset R^N$, $N = 1, 2, 3$, and for $N > 1$ it is assumed that the boundary $\partial\Omega$ of $\Omega$ is sufficiently smooth.

When $\alpha = 1$ the equation in (1) is also called the Haller equation. This equation is a mathematical model of water movement in capillary-porous media, including the soil. In this case, $u$ is humidity in fractions of a unit, $x$ is a point inside the soil and $t$ is time (see, for example, in [5–7]). See also [8,9], where on the base of the modified Darcy's law for a viscoelastic fluid, the first Stokes problem was extended to the problem for an Oldroyd-B fluid in a porous half-space, and Equation (1) was obtained as a mathematical model. Recall that usually Stokes' first problem describes flows caused by a suddenly accelerated plate for homogeneous incompressible isotropic fluids with pressure-dependent viscosity.

The Rayleigh–Stokes problem (1) plays an important role in the study of the behavior of some non-Newtonian fluids as well. A non-Newtonian fluid is a fluid that has a constant viscosity independent of stress, i.e., does not obey Newton's law of viscosity. The fractional derivative $\partial_t^\alpha$ is used in Equation (1) to describe the viscoelastic flow behavior (see, for example, [10,11]).

In recent years, a number of papers have been devoted to the Rayleigh–Stokes problem (1) because of its importance for applications (see, for example, [8–18]). An overview of work in this direction can be found in Bazhlekova et al. [4] (see also [18]). The properties of the solution of this model were studied by a number of authors applying various methods; see, e.g., [10–12]. The authors of the work by Bazhlekova et al. [4] proved the Sobolev regularity of the homogeneous Rayleigh–Stokes problem for both smooth and non-smooth initial data $\varphi(x)$, including $\varphi(x) \in L_2(\Omega)$. A number of authors have studied efficient and accurate numerical algorithms for solving problem (1). A survey of works in this direction is contained in the above-mentioned paper [4]. See also the recent articles [13,14] and references therein.

The study of the inverse problem of determining the right-hand side of the Rayleigh–Stokes equation is the subject of many studies (see, for example, [15–17] and the bibliography cited there). Since this inverse problem is ill-posed in the sense of Hadamard, various regularization methods are considered in the above studies, and numerical methods for finding the right-hand side of the equation are also proposed. We note also that the inverse problem of determining the right-hand side of the equation is also ill-posed for the subdiffusion equation (see, for example, [19–21]).

If in problem (1) we replace the initial condition $u(x,0) = \varphi(x)$ by $u(x,T) = \varphi(x)$, then we obtain the so-called backward problem. This problem is ill-posed, since a small change in the current state $u(x,T)$ leads to a large change in the solution. In the papers [22,23] (see also references therein) various regularization methods are proposed, accompanied by verification of these methods using numerical experiments. We emphasize that in these papers $N < 4$, and it has to do with the method used there. Namely, if the dimension of the domain $\Omega$ is less than four, then the series

$$\sum_k \lambda_k^{-2},$$

composed of the eigenvalues $\lambda_k$ of the Laplace operator with the Dirichlet condition converges.

Let us focus, in more detail, on the recently published study [18]. In this paper, along with other questions, problem (1) is investigated by taking the non-local condition $u(x,T) = \beta u(x,0) + \varphi(x)$ instead of the initial condition. The authors considered only the cases $\beta = 0$ and $\beta = 1$: if $\beta = 0$ then we have the backward problem (note that here the dimension $N$ is arbitrary). The authors proved that if $\beta = 0$, then the solution exists and is unique, but there is no stability. If $\beta = 1$, then the problem is well-posed in the sense

of Hadamard, i.e., a unique solution exists and the solution continuously depends on the initial data and on the right-hand side of the equation.

The question naturally arises: what happens if $\beta$ takes other values than 0 and 1? In the present paper we consider a more general non-local condition $u(x, t_0) = \beta u(x, 0) + \varphi(x)$, $t_0 \in (0, T]$ and provide a definitive answer to this question. The main results of the current work can be formulated as follows:

(1)  If $\beta \geq 1$ or $\beta < 0$, then the problem is well-posed in the sense of Hadamard: the solution exists and unique and stable;
(2)  The case $\beta = 0$ is considered in [18]: in this case there is a unique solution, but it is not stable;
(3)  If $\beta \in (0, 1)$, then the well-posedness of the problem depends on the location of the spectrum (i.e., the eigenvalues $\lambda_k$) of the Laplace operator with the Direchlet condition. If the inequality $B_\alpha(\lambda_k, t_0) \neq \beta$ (the definition of this function is given in Section 3) holds for all $k = 1, 2, \cdots$, then the problem is well-posed in the sense of Hadamard. If $B_\alpha(\lambda_k, t_0) = \beta$ for some $k \in K_0$ (it is proved in the paper that the set $K_0$ contains only a finite number of points), then a necessary and sufficient condition for the existence of a solution is found. However, in this case there is no unique solution.

In what concerns the non-local condition

$$u(x, T) = \beta u(x, 0) + \varphi(x),$$

in the variable $t$, the corresponding problem with the parameter $\beta = 1$ for the classical diffusion equation was first considered in [24–26]. For subdiffusion equations with the Riemann–Liouville and the Caputo derivatives; this problem was studied in detail in the papers [27,28], respectively. It should be emphasized that the parameter $\beta$ in these papers is an arbitrary real number. The authors of a recent paper [29] investigated the subdiffusion equation with the Caputo–Fabrizio derivative on an $N$-dimensional torus with the non-local condition

$$\varepsilon u(T) = u(0) + \varphi.$$

In these studies the cases $\varepsilon = 0$ and $\varepsilon > 0$ are studied separately. The authors also studied the solution limit at $\varepsilon \to 0$. Note, in this paper if $\varepsilon = 0$, then we have the Cauchy problem, whereas in our case we have the backward problem.

The present paper consists of five sections. Section 2 provides precise formulations of the problems studied in this paper. In Section 3, we introduce the standard Hilbert space of "smooth" functions via the power of an elliptic operator and give some well-known properties of the function $B_\alpha(\lambda, t)$ introduced in [4]. Here we prove an important lemma used for the solution of the non-local problem in the variable $t$. Section 4 is devoted to the main result of this paper, where a non-local problem with an operator $A$ generalizing the Laplace operator is studied.

## 2. Problem Formulations

Let $H$ be a separable Hilbert space. Denote by $(\cdot, \cdot)$ be the inner product and by $|| \cdot ||$ the norm in $H$. Consider an arbitrary unbounded positive self-adjoint operator $A$ with a dense domain in $H$. We assume that $A$ has a complete in $H$ system of orthonormal eigenvectors (eigenfunctions) $\{v_k\}$ and a countable set of positive eigenvalues

$$\lambda_k : \quad 0 < \lambda_1 \leq \lambda_2 \cdots \to +\infty.$$

We also assume that the set $\{\lambda_k\}$ does not have a finite limit point.

For a vector-valued functions $h : \mathbb{R}_+ \to H$, we define the Riemann–Liouville fractional derivative of order $0 < \alpha < 1$ in the same way as (2) (see, e.g., [30]). Finally, let $C((a, b); H)$ denote the set of functions $u(t)$ continuous in $t \in (a, b)$ with values in $H$.

Consider the following non-local problem for the abstract Rayleigh–Stokes equation accepting the integral (in the definition of the fractional derivative) in the sense of Bochner:

$$\begin{cases} \partial_t u(t) + (1 + \gamma\, \partial_t^\alpha) A u(t) = f(t), & 0 < t \le T; \\ u(t_0) = \beta u(0) + \varphi, \end{cases} \tag{3}$$

where $\gamma > 0$ and $t_0 \in (0, T]$ are fixed constants, $\varphi \in H$, $f(t) \in C((0, T]; H)$ and $\beta$ is an arbitrary fixed real number. If $\beta = 0$, then this problem is called *the backward problem*.

**Definition 1.** *If a function $u(t) \in C([0, T]; H)$ satisfies the conditions*

$$\partial_t u(t),\ A u(t),\ \partial_t^\alpha A u(t) \in C((0, T); H),$$

*and (3) for all $t \in (0, T]$, then it is called a solution of the non-local Rayleigh–Stokes problem (3).*

**Remark 1.** *Let $\Omega$ be an arbitrary bounded N-dimensional domain (N is not necessarily $\le 3$) Then, as the operator A, we can take the Laplace operator with the Dirichlet condition in this domain. This operator has all the properties listed above.*

## 3. Preliminaries

For a given real number $\tau$, we define the operator $A^\tau$ by

$$A^\tau h = \sum_{k=1}^\infty \lambda_k^\tau h_k v_k,$$

with the domain of definition

$$D(A^\tau) = \{h \in H : \sum_{k=1}^\infty \lambda_k^{2\tau} |h_k|^2 < \infty\}.$$

Note that $\lambda_k > 0$ for all $k$, which is a consequence of the fact that the operator $A$ is positive. Here and below, the symbols $h_k$, $k = 1, 2, \ldots$, denote the Fourier coefficients of the vector $h \in H$: $h_k = (h, v_k)$.

For elements of $h, g \in D(A^\tau)$ we introduce the norm

$$||h||_\tau^2 = \sum_{k=1}^\infty \lambda_k^{2\tau} |h_k|^2 = ||A^\tau h||^2,$$

and the inner product

$$(h, g)_\tau = \sum_{k=1}^\infty \lambda_k^{2\tau} h_k g_k = (A^\tau h, A^\tau g).$$

With this inner product, the linear-vector space $D(A^\tau)$ becomes a Hilbert space. Further, let $B_\alpha(\lambda, t)$ be a solution of the following Cauchy problem

$$L y(t) \equiv y'(t) + \lambda(1 + \gamma \partial_t^\alpha) y(t) = 0,\ t > 0,\ \lambda > 0,\ y(0) = 1.$$

$B_\alpha(\lambda, t)$ can be expressed in terms of the generalized Wright function (see, e.g., [1,31]). The properties of $B_\alpha(\lambda, t)$ is studied in detail in Bazhlekova et al. [4]. See also Luc et al. [22], where important lower bounds are obtained. The authors of [4], in particular, proved the following lemma.

**Lemma 1.** *The following statements are true:*

*1.*

$$B_\alpha(\lambda, 0) = 1,\ 0 < B_\alpha(\lambda, t) < 1,\ t > 0, \tag{4}$$

2.  $\lambda B_\alpha(\lambda, t) < C \min\{t^{-1}, t^{\alpha-1}\}, \ t > 0.$

The function $B_\alpha(\lambda, t)$ has the representation [4]

$$B_\alpha(\lambda, t) = \int_0^\infty e^{-rt} b_\alpha(\lambda, r) dr, \tag{5}$$

where

$$b_\alpha(\lambda, r) = \frac{\gamma}{\pi} \frac{\lambda r^\alpha \sin \alpha \pi}{(-r + \lambda \gamma r^\alpha \cos \alpha \pi + \lambda)^2 + (\lambda \gamma r^\alpha \sin \alpha \pi)^2}. \tag{6}$$

**Lemma 2** ([4,18])**.** *The Cauchy problem*

$$y'(t) + \lambda(1 + \gamma \partial_t^\alpha) y(t) = f(t), \ t > 0, \ \lambda > 0, \ y(0) = y_0,$$

*has a unique solution, which has a representation*

$$y(t) = y_0 B_\alpha(\lambda, t) + \int_0^t B_\alpha(\lambda, t - \tau) f(\tau) d\tau.$$

We will also need an estimate obtained in [18] for the derivative of the function $B_\alpha(\lambda, t)$. In view of the importance of this assertion for our further considerations, we present it with a brief proof.

**Lemma 3.** *There is a constant $C > 0$ such that*

$$|\partial_t B_\alpha(\lambda, t)| \le \frac{C}{\lambda \, t^{2-\alpha}}, \ t > 0.$$

**Proof.** Differentiating the function $B_\alpha(\lambda, t)$ defined in (5), we have

$$\partial_t B_\alpha(\lambda, t) = -\int_0^\infty r e^{-rt} b_\alpha(\lambda, r) dr.$$

Therefore, in accordance with the definition of $b_\alpha(\lambda, r)$ in (6)

$$|\partial_t B_\alpha(\lambda, t)| \le \frac{\gamma}{\pi} \int_0^\infty \frac{\lambda r^\alpha \sin \alpha \pi}{(\lambda \gamma r^\alpha \sin \alpha \pi)^2} r e^{-rt} dr$$

$$= \frac{1}{\gamma \pi \lambda \sin \alpha \pi} \int_0^\infty r^{1-\alpha} e^{-rt} dr.$$

Now the change in the variable $\tau = rt$ implies

$$|\partial_t B_\alpha(\lambda, t)| \le \frac{t^{\alpha-2}}{\gamma \pi \lambda \sin \alpha \pi} \int_0^\infty \tau^{1-\alpha} e^{-\tau} d\tau$$

$$= \frac{t^{\alpha-2}}{\gamma \pi \lambda \sin \alpha \pi} \Gamma(2 - \alpha) = \frac{C}{\lambda \, t^{2-\alpha}}.$$

$\square$

In what follows, $\lambda$ will be replaced by the eigenvalues $\lambda_k$ of the operator $A$. The following important lower bound for $B_\alpha(\lambda_k, t)$ was obtained in Luc, N.H., Tuan, N.H., Kirane, M., Thanh, D.D.X. [22].

**Lemma 4.** *For all $t \in [0, T]$ and $k \geq 1$ one has:*

$$B_\alpha(\lambda_k, t) \geq \frac{C(\alpha, \gamma, \lambda_1)}{\lambda_k},$$

*where*

$$C(\alpha, \gamma, \lambda_1) = \frac{\gamma \sin \alpha \pi}{4} \int_0^\infty \frac{r^\alpha e^{-rT}}{\frac{r^2}{\lambda_1^2} + \gamma^2 r^{2\alpha} + 1} dr.$$

Next, we estimate the derivative $\partial_\lambda B_\alpha(\lambda, t_0)$ from above when $\lambda \geq \lambda_1 > 0$, where $\lambda_1$ is the first eigenvalue of the operator $A$.

**Lemma 5.** *Let $0 < t_0 \leq T$, $\gamma > 0$, $\alpha \in (0, 1)$ be given numbers. There exists a positive number $\Lambda_0 = \Lambda_0(t_0, \gamma, \alpha, \lambda_1) > 0$ such that for any $\lambda \geq \Lambda_0$ the inequality*

$$\partial_\lambda B_\alpha(\lambda, t_0) < 0$$

*holds.*

**Proof.** We rewrite the function $B_\alpha(\lambda, t_0)$ in the form

$$B_\alpha(\lambda, t_0) = \frac{1}{\lambda} \int_0^\infty e^{-t_0 r} b_{\alpha,1}(\lambda, r) dr,$$

where

$$b_{\alpha,1}(\lambda, r) = \frac{\gamma}{\pi} \frac{r^\alpha \sin \alpha \pi}{(-\frac{r}{\lambda} + \gamma r^\alpha \cos \alpha \pi + 1)^2 + (\gamma r^\alpha \sin \alpha \pi)^2}.$$

Now, differentiating with respect to $\lambda$ we have

$$\partial_\lambda B_\alpha(\lambda, t_0) = -\frac{1}{\lambda^2} \int_0^\infty e^{-t_0 r} b_{\alpha,1}(\lambda, r) dr$$

$$+ \frac{2}{\lambda^3} \int_0^\infty e^{-t_0 r} b_{\alpha,1}(\lambda, r) \frac{r[\frac{r}{\lambda} - \gamma r^\alpha \cos \alpha \pi - 1]}{(-\frac{r}{\lambda} + \gamma r^\alpha \cos \alpha \pi + 1)^2 + (\gamma r^\alpha \sin \alpha \pi)^2} dr. \quad (7)$$

We estimate each term in the latter separately. For the first integral, taking into account the inequality $(a + b + c)^2 \leq 3(a^2 + b^2 + c^2)$, we have

$$(-\frac{r}{\lambda} + \gamma r^\alpha \cos \alpha \pi + 1)^2 + (\gamma r^\alpha \sin \alpha \pi)^2 \leq 3(\frac{r^2}{\lambda_1^2} + \gamma^2 r^{2\alpha} + 1) + \gamma^2 r^{2\alpha}$$

$$\leq \begin{cases} c, & r < 1; \\ c r^2, & r \geq 1. \end{cases} \quad (8)$$

We note that here and elsewhere in this section, $c$ denotes a constant (not necessarily the same one), depending on the fixed parameters $\lambda_1$, $\alpha$ and $\gamma$. Therefore,

$$\int_0^1 e^{-t_0 r} b_{\alpha,1}(\lambda, r) dr \geq c \int_0^1 r^\alpha e^{-rt_0} dr = c t_0^{-1-\alpha} I_1(t_0), \quad (9)$$

where

$$I_1(t_0) = \int_0^{t_0} \xi^\alpha e^{-\xi} d\xi, \quad (10)$$

and

$$\int\limits_1^\infty e^{-t_0 r} b_{\alpha,1}(\lambda, r) dr \geq c \int\limits_1^\infty r^{\alpha-2} e^{-r t_0} dr = c\, t_0^{1-\alpha}\, I_2(t_0) \tag{11}$$

where

$$I_2(t_0) = \int\limits_{t_0}^\infty \zeta^{\alpha-2} e^{-\zeta} d\zeta. \tag{12}$$

It follows from estimates (9) and (11) that

$$-\frac{1}{\lambda^2} \int\limits_0^\infty e^{-t_0 r} b_{\alpha,1}(\lambda, r) dr \leq -\frac{C_0(t_0)}{\lambda^2}, \tag{13}$$

where $C_0(t_0) = c\left(t_0^{-1-\alpha} I_1(t_0) + t_0^{1-\alpha} I_2(t_0)\right)$.

Let us now estimate the second term in (7) from above. Denote

$$R(r) = -\frac{r}{\lambda} + \gamma r^\alpha \cos \alpha \pi + 1.$$

Then $R(0) = 1$ and we can choose a positive number $r_0$ such that for all $r \in (0, r_0)$ one has $R(r) \geq \varepsilon$ with some $\varepsilon > 0$. For these $r$ we have

$$b_{\alpha,1}(\lambda, r) \leq \frac{\gamma}{\pi}\, \frac{r^\alpha \sin \alpha \pi}{(-\frac{r}{\lambda} + \gamma r^\alpha \cos \alpha \pi + 1)^2} \leq \frac{\gamma}{\varepsilon^2 \pi}\, r^\alpha \sin \alpha \pi.$$

Taking this into account, we obtain

$$\int\limits_0^{r_0} e^{-t_0 r} b_{\alpha,1}(\lambda, r)\, \frac{r\left[\frac{r}{\lambda} - \gamma r^\alpha \cos \alpha \pi - 1\right]}{(-\frac{r}{\lambda} + \gamma r^\alpha \cos \alpha \pi + 1)^2 + (\gamma r^\alpha \sin \alpha \pi)^2} dr$$

$$\leq \frac{\gamma}{\varepsilon^3 \pi} \sin \alpha \pi \int\limits_0^{r_0} e^{-t_0 r} r^{\alpha+1} dr$$

$$= \frac{c}{t_0^{\alpha+2}} \int\limits_0^{r_0 t_0} e^{-\zeta} \zeta^{\alpha+1} dr \leq \frac{c\,\Gamma(\alpha+2)}{t_0^{\alpha+2}}. \tag{14}$$

Now let $r \geq r_0$. Then, evidently,

$$b_{\alpha,1}(\lambda, r) \leq \frac{1}{\gamma \pi}\, \frac{1}{r^\alpha \sin \alpha \pi},$$

and, moreover,

$$\frac{r\left|\frac{r}{\lambda} - \gamma r^\alpha \cos \alpha \pi - 1\right|}{(-\frac{r}{\lambda} + \gamma r^\alpha \cos \alpha \pi + 1)^2 + (\gamma r^\alpha \sin \alpha \pi)^2}$$

$$\leq \frac{r^{1-2\alpha}}{(\gamma \sin \alpha \pi)^2}\left|\frac{r}{\lambda} - \gamma r^\alpha \cos \alpha \pi - 1\right| \leq \frac{C r^{2-2\alpha}}{(\gamma \sin \alpha \pi)^2},$$

where $C = \lambda_1^{-1} + r_0^{\alpha-1}\gamma^{-1} + r_0^{-1}$. Hence,

$$\int_{r_0}^{\infty} e^{-t_0 r} b_{\alpha,1}(\lambda, r) \frac{r\left|\frac{r}{\lambda} - \gamma r^{\alpha} \cos \alpha \pi - 1\right|}{\left(-\frac{r}{\lambda} + \gamma r^{\alpha} \cos \alpha \pi + 1\right)^2 + (\gamma r^{\alpha} \sin \alpha \pi)^2} dr$$

$$\leq \frac{C}{\pi(\gamma \sin \alpha \pi)^3} \int_{r_0}^{\infty} e^{-t_0 r} r^{2-3\alpha} dr \leq \frac{C}{\pi(\gamma r_0^{\alpha} \sin \alpha \pi)^3} \int_{0}^{\infty} e^{-t_0 r} r^2 dr$$

$$= \frac{C\Gamma(3)}{\pi(\gamma r_0^{\alpha} \sin \alpha \pi)^3} \cdot \frac{1}{t_0^3}.$$

It follows from the latter and estimate (14) that for all $\lambda \geq \lambda_1$ the second term in (7) is estimated from above by the quantity

$$\frac{c}{\lambda^3} \cdot \left[\frac{1}{t_0^{\alpha+2}} + \frac{1}{t_0^3}\right]. \tag{15}$$

Finally, estimates (13) and (15) imply

$$\partial_{\lambda} B_{\alpha}(\lambda, t_0) \leq -\frac{1}{\lambda^2}\left[C_0(t_0) - \frac{c}{\lambda}\left(\frac{1}{t_0^{\alpha+2}} + \frac{1}{t_0^3}\right)\right] < 0, \quad \lambda \geq \lambda_1. \tag{16}$$

Therefore, for sufficiently large $\lambda$, depending on $t_0, \gamma, \alpha$ and $\lambda_1$, this implies the assertion of the lemma. $\square$

Let $\{\lambda_k\}$ be the set of eigenvalues of the operator $A$. Recall that this set has no finite limit points. In particular, the multiplicity of any eigenvalue is finite. Let $\beta \in (0,1)$. In our further analysis of non-local problem (3) we will encounter the solution of the following equation

$$B_{\alpha}(\lambda, t_0) = \beta \tag{17}$$

with respect to $\lambda$. If $\lambda_0$ is a root of Equation (17), then the set of all $k$ for which $\lambda_k = \lambda_0$ will be denoted by $K_0$. If there is not an eigenvalue $\lambda_k$ equal to $\lambda_0$, then evidently the set $K_0$ is empty.

**Remark 2.** *According to Lemma 5, starting from some number k, the function $B_{\alpha}(\lambda_k, t_0)$ decreases strictly with respect to $\lambda_k$ (if the multiplicity of the eigenvalue $\lambda_k$ is not taken into account). Therefore, the set $K_0$ is always finite.*

Thus, Lemma 5 states that only a finite number of eigenvalues $\lambda_k$ can be solutions of Equation (17). It can also be proved that if $t_0$ is large enough, then there can be only one such eigenvalue. Indeed, the following statement is true:

**Lemma 6.** *Let $\gamma > 0$ and $\alpha \in (0,1)$ be given numbers. There exists a positive number $T_0 = T_0(\gamma, \alpha, \lambda_1) \geq 1$ such that for any $t_0$ in the interval $T_0 \leq t_0 \leq T$ the inequality*

$$\partial_{\lambda} B_{\alpha}(\lambda, t_0) < 0, \quad \lambda \geq \lambda_1.$$

*holds.*

**Proof.** Let us write estimate (16) for $\lambda = \lambda_1$:

$$\partial_{\lambda} B_{\alpha}(\lambda, t_0) \leq -\frac{1}{\lambda_1^2}\left[C_0(t_0) - \frac{c}{\lambda_1}\left(\frac{1}{t_0^{\alpha+2}} + \frac{1}{t_0^3}\right)\right]. \tag{18}$$

Now suppose that $t_0 \geq 1$ and estimate $C_0(t_0)$ from below. For the integral $I_1(t_0)$, defined in (10), we have

$$I_1(t_0) \geq \int_0^1 \xi^\alpha e^{-\xi} d\xi \geq \frac{1}{(\alpha + 1) e}.$$

Similarly, for the integral $I_2(t_0)$, defined in (12), after integration by parts twice, we obtain

$$I_2(t_0) = \frac{t_0^{\alpha-1} e^{-t_0}}{1 - \alpha} + \frac{t_0^\alpha e^{-t_0}}{(1 - \alpha)\alpha} + \frac{1}{(1 - \alpha)\alpha} \int_{t_0}^\infty \xi^\alpha e^{-\xi} d\xi$$

$$\geq e^{-t_0} \left[ \frac{t_0^{\alpha-1}}{1 - \alpha} + \frac{t_0^\alpha}{(1 - \alpha)\alpha} + \frac{t_0^{\alpha+1}}{(1 - \alpha)\alpha(\alpha + 1)} \right].$$

Therefore, for sufficiently large $t_0$ we have

$$C_0(t_0) \geq c\, t_0^{-1-\alpha}.$$

Consequently, estimate (18) takes the form

$$\partial_\lambda B_\alpha(\lambda, t_0) \leq -\frac{c}{\lambda_1^2 t_0^{1+\alpha}} \left[ 1 - \frac{1}{\lambda_1} \left( \frac{1}{t_0} + \frac{1}{t_0^{2-\alpha}} \right) \right].$$

This implies the assertion of the lemma. $\square$

**Remark 3.** *Under the conditions of this lemma, for all $t_0 \in [T_0, T]$ only one eigenvalue $\lambda_{k_0}$ may satisfy Equation (17). Let the multiplicity of $\lambda_{k_0}$ be equal to $p$. Then $K_0 = \{k_0, k_0 + 1, \cdots, k_0 + p - 1\}$.*
*We also note that in Lemma 5 $\lambda$ is sufficiently large and $t_0$ is an arbitrary positive number; in Lemma 6, on the contrary, $t_0$ is sufficiently large and $\lambda \geq \lambda_1$ is an arbitrary number.*

## 4. Existence of a Solution of the Non-Local Problem (3)

To solve the non-local problem (3), we divide it into two auxiliary problems:

$$\begin{cases} \partial_t \omega(t) + (1 + \gamma\, \partial_t^\alpha) A\omega(t) = f(t), & 0 < t \leq T, \\ \omega(0) = 0, \end{cases} \tag{19}$$

and

$$\begin{cases} \partial_t w(t) + (1 + \gamma\, \partial_t^\alpha) Aw(t) = 0, & 0 < t \leq T, \\ w(t_0) = \beta w(0) + \psi, \end{cases} \tag{20}$$

where $\psi \in H$ is a given element, $t_0$ is any fixed number from $(0, T]$ and $\beta$ is a fixed real number.

Since problems (19) and (20) are special cases of problem (3), the solutions to problems (19) and (20) are determined completely similarly to Definition 1.

**Lemma 7.** *Let $\psi$ in (20) have the form $\psi = \varphi - \omega(t_0)$, where $\varphi$ is a function in non-local problem (3). Then the solution to problem (3) has the form $u(t) = \omega(t) + w(t)$, where $\omega(t)$ and $w(t)$ are solutions of problems (19) and (20), respectively.*

**Proof.** Put the function $u(t) = \omega(t) + w(t)$ in Equation (3). Then due to Equations (19) and (20) one has

$$\partial_t(\omega(t) + w(t)) + (1 + \gamma\, \partial_t^\alpha) A(\omega(t) + w(t)) = f(t).$$

Now let us check the validity of the non-local condition in problem (3):

$$u(t_0) = \omega(t_0) + w(t_0) = \beta\omega(0) + \beta w(0) + \varphi.$$

Using the Cauchy condition for $\omega(t)$ and the non-local condition for $w(t)$, we obtain (note, $\psi = \varphi - \omega(t_0)$)

$$\omega(t_0) = \beta w(0) - w(t_0) + \varphi = -\psi + \varphi = \omega(t_0),$$

and this identity proves the lemma. □

Auxiliary problem (19) is solved in [18]. Let us formulate the corresponding result:

**Theorem 1.** *Let $f(t) \in C([0, T]; D(A^\varepsilon))$ for some $\varepsilon \in (0, 1)$. Then the Cauchy problem (19) has a unique solution*

$$\omega(t) = \sum_{k=1}^{\infty} \left[ \int_0^t B_\alpha(\lambda_k, t - \tau) f_k(\tau) d\tau \right] v_k.$$

*Moreover, the following estimate holds*

$$||\partial_t \omega(t)||^2 + ||\partial_t^\alpha A \omega(t)||^2 \leq C_\varepsilon \max_{t \in [0,T]} ||f||_\varepsilon^2, \quad C_\varepsilon > 0, \quad 0 \leq t \leq T.$$

Now let us consider non-local problem (20). We will seek the solution of this problem in the form of a generalized Fourier series

$$w(t) = \sum_{k=1}^{\infty} T_k(t) v_k,$$

where $v_k$ are the eigenvectors of the operator $A$ and $T_k(t)$, $k \geq 1$, are solutions of the following non-local problems:

$$\begin{cases} T_k'(t) + \lambda_k (1 + \gamma \partial_t^\alpha) T_k(t) = 0, & 0 < t \leq T; \\ T_k(t_0) = \beta T_k(0) + \psi_k, \end{cases} \tag{21}$$

where $k \geq 1$, $t_0 \in (0, T]$ is a fixed point and $\psi_k$ is the Fourier coefficient of the element $\psi \in H$. Denote $h_k = T_k(0), k = 1, 2, \ldots$. Then the unique solution to problem (21) has the form (see Lemma 2)

$$T_k(t) = h_k B_\alpha(\lambda_k, t).$$

To find the unknown numbers $h_k$, we use the non-local conditions of (21). Namely,

$$h_k B_\alpha(\lambda_k, t_0) = \beta h_k + \psi_k,$$

or, equally,

$$h_k (B_\alpha(\lambda_k, t_0) - \beta) = \psi_k. \tag{22}$$

If $\beta \geq 1$ or $\beta < 0$ (note, $t_0 > 0$ and $\lambda_k > 0$), then $B_\alpha(\lambda_k, t_0) \neq \beta$ due to Lemma 1. Therefore, in these cases it follows from (22) that

$$h_k = \frac{\psi_k}{B_\alpha(\lambda_k, t_0) - \beta} \tag{23}$$

and

$$|h_k| \leq C_\beta |\psi_k|, \quad k \geq 1, \tag{24}$$

where $C_\beta$ is a constant depending on $\beta$. If $\beta = 0$, then $B_\alpha(\lambda_k, t_0) \neq 0$; however, in accordance with Lemma 1, the function $B_\alpha(\lambda_k, t_0)$ asymptotically tends to zero as $k \to \infty$. Therefore, by Lemma 4, one has:

$$h_k = \frac{\psi_k}{B_\alpha(\lambda_k, t_0)}, \quad C_1 \lambda_k |\psi_k| \leq |h_k| \leq C_2 \lambda_k |\psi_k|.$$

Here, the constants $C_j, j = 1, 2$, may depend on $\alpha, \gamma, \lambda_1$ and $t_0$. As noted above, this case has been studied in detail in [18]. Therefore, we will not consider it further.

Now, let $0 < \beta < 1$ and consider Equation (17). In accordance with Remark 2, there are two possible cases: the set $K_0$ is empty or it is not empty. If $K_0$ is empty, then since the set $\{\lambda_k\}$ does not have a finite limit point, the estimate in (24) holds with some constant $C_\beta > 0$ for all $k$.

Thus, if $\beta \in (0, 1)$ and $K_0$ is empty, then the formal solution of problem (20) still has the form

$$w(t) = \sum_{k=1}^{\infty} \frac{\psi_k}{B_\alpha(\lambda_k, t_0) - \beta} B_\alpha(\lambda_k, t) v_k. \tag{25}$$

Finally, let us assume that $0 < \beta < 1$ and the set $K_0$ is not empty. In this case, due to Equation (22), non-local problem (21) has a solution if and only if the following orthogonality conditions are verified:

$$\psi_k = (\psi, v_k) = 0, \ k \in K_0. \tag{26}$$

Moreover, for the values $k \in K_0$ arbitrary numbers $h_k$ are solutions of Equation (22). For all other $k$ we have

$$h_k = \frac{\psi_k}{B_\alpha(\lambda_k, t_0) - \beta}, \quad |h_k| \le C_\beta |\psi_k|, \quad k \notin K_0. \tag{27}$$

Thus, the formal solution of problem (20) in this case has the form

$$w(t) = \sum_{k \notin K_0} \frac{\psi_k}{B_\alpha(\lambda_k, t_0) - \beta} B_\alpha(\lambda_k, t) v_k + \sum_{k \in K_0} h_k B_\alpha(\lambda_k, t) v_k. \tag{28}$$

Now let us show that the series (25) and (28) indeed define solutions to non-local problem (20). According to Definition 1, it suffices to show the applicability of the operators $\partial_t$ and $\partial_t^\alpha A$ term-by-term to these series and $w(t) \in C([0, T]; H)$, $\partial_t w(t), Aw(t)$, $\partial_t^\alpha Aw(t) \in C((0, T); H)$. We demonstrate this with the solution in (25). Concerning the solution in (28), it is treated in exactly the same way.

Let $S_j(t), j \ge 1$, be the sequence of partial sums of series (25). Applying Parseval's equality, estimate (4), and the first assertion of Lemma 1, we have

$$||S_j(t)||^2 = \sum_{k=1}^{j} \left| \frac{\psi_k}{B_\alpha(\lambda_k, t_0) - \beta} B_\alpha(\lambda_k, t) \right|^2 \le C_\beta ||\psi||^2.$$

Letting $j \to \infty$, it follows from the latter that $w(t) \in C([0, T]; H)$. Further, we have

$$AS_j(t) = \sum_{k=1}^{j} \frac{\lambda_k \psi_k}{B_\alpha(\lambda_k, t_0) - \beta} B_\alpha(\lambda_k, t) v_k.$$

Using the same reasoning (using the third assertion of Lemma 1) as above, we obtain

$$||AS_j(t)||^2 = \sum_{k=1}^{j} \left| \frac{\lambda_k \psi_k}{B_\alpha(\lambda_k, t_0) - \beta} B_\alpha(\lambda_k, t) \right|^2 \le \frac{C_\beta}{t^{2(1-\alpha)}} ||\psi||^2, \tag{29}$$

which implies that $Aw(t) \in C((0, T); H)$. Next, applying Lemma 3, we have the following estimate:

$$||\partial_t S_j(t)||^2 = \sum_{k=1}^{j} \left| \frac{\psi_k}{B_\alpha(\lambda_k, t_0) - \beta} \partial_t B_\alpha(\lambda_k, t) \right|^2 \le \frac{C_\beta}{\lambda_1 t^{2(2-\alpha)}} ||\psi||^2. \tag{30}$$

The latter implies $\partial_t w(t) \in C((0, T]; H)$. Equation (20) written in the form $\partial_t w(t) = -Aw(t) - \gamma \partial_t^\alpha Aw(t)$, $t > 0$, and the estimates obtained above imply

$$||\partial_t^\alpha Aw(t)||^2 \leq \frac{C_\beta}{t^{2(2-\alpha)}}||\psi||^2. \tag{31}$$

Hence, $\partial_t^\alpha Aw(t) \in C((0, T]; H)$ as well.

Thus, if $\beta \notin (0, 1)$ or $\beta \in (0, 1)$, but $K_0$ is an empty set, then the function (25) is indeed a solution to problem (20).

To prove the uniqueness of the solution to problem (20), it suffices to show that the solution of the homogeneous problem

$$\begin{cases} \partial_t w(t) + (1 + \gamma \partial_t^\alpha) Aw(t) = 0, & 0 < t \leq T; \\ w(t_0) = \beta w(0), \end{cases}$$

is identically zero: $w(t) \equiv 0$.

Let $w(t)$ be any solution to this problem and let $w_k(t) = (w(t), v_k)$. Since the operator $A$ is self-adjoint, one has

$$\begin{aligned} \partial_t^\rho w_k(t) = (\partial_t^\rho w(t), v_k) &= -(Aw(t), v_k) - \gamma(\partial_t^\alpha w, v_k) \\ &= -\lambda_k (1 + \gamma \partial_t^\alpha) w_k(t) \end{aligned}$$

or

$$\partial_t^\rho w_k(t) + \lambda_k (1 + \gamma \partial_t^\alpha) w_k(t) = 0. \tag{32}$$

It follows from the nonlocal condition that

$$w_k(t_0) = \beta w_k(0). \tag{33}$$

Let us denote $w_k(0) = h_k$. Then the unique solution to the differential Equation (32) with this initial condition has the form $w_k(t) = h_k B_\alpha(\lambda_k, t)$ (see Lemma 2). Using condition (33) we obtain the following equation for unknown numbers $h_k$:

$$h_k B_\alpha(\lambda_k, t_0) = \beta h_k. \tag{34}$$

Let $\beta \notin (0, 1)$ or $\beta \in (0, 1)$, but $K_0$ be an empty set. Then $B_\alpha(\lambda_k, t) \neq \beta$ for all $k$. Consequently, in this case all $h_k$ are equal to zero. Therefore $w_k(t) = 0$, and by virtue of the completeness of the set of eigenfunctions $\{v_k\}$, we conclude that $w(t) \equiv 0$. Thus, problem (20) in this case has a unique solution.

Now consider the case $\beta \in (0, 1)$ and $K_0$ not empty. Then $B_\alpha(\lambda_k, t) = \beta$, $k \in K_0$ and therefore, Equation (34) has the following solution: $h_k = 0$ if $k \notin K_0$ and $h_k$ is an arbitrary number if $k \in K_0$. Thus, in this case, there is no uniqueness of the solution to problem (20). We note that the non-local problem under consideration has a finite-dimensional kernel

$$Ker = \{h \in H : h = \sum_{k \in K_0} h_k v_k\}$$

in this case.

Thus, we obtain the following statement:

**Theorem 2.** *Let $\psi \in H$. If $\beta \notin [0, 1)$ or $\beta \in (0, 1)$, but $K_0$ is empty, then problem (20) has a unique solution and this solution has the form (25). If $\beta \in (0, 1)$ and $K_0$ is not empty, then a solution to problem (20) exists if and only if the orthogonality conditions (26) are satisfied. The solution of problem (20) has the form (28) with arbitrary coefficients $h_k$, $k \in K_0$. Moreover, there is a constant $C_\beta > 0$ such that the following coercive estimate holds:*

$$||\partial_t w(t)||^2 + ||Aw(t)||^2 + ||\partial_t^\alpha Aw(t)||^2 \leq C_\beta t^{-2(1-\alpha)}||\psi||^2, \quad 0 < t \leq T. \tag{35}$$

Note that the proof of the coercive inequality (35) follows from estimates (29)–(31).

Now we are ready to solve the main problem in (3). Let $\varphi \in H$ and $f(t) \in C([0, T]; D(A^\varepsilon))$ for some $\varepsilon \in (0, 1)$. If we put $\psi = \varphi - \omega(t_0) \in H$ and $\omega(t)$ and $w(t)$ are the corresponding solutions of problems (19) and (20), then the function $u(t) = \omega(t) + w(t)$ is a solution to problem (3). Therefore, if $\beta \notin (0, 1)$ or $\beta \in (0, 1)$, but $K_0$ is empty, then

$$u(t) = \sum_{k=1}^{\infty} \left[ \frac{\varphi_k - \omega_k(t_0)}{B_\alpha(\lambda_k, t_0) - \beta} B_\alpha(\lambda_k, t) + \omega_k(t) \right] v_k, \tag{36}$$

where

$$\omega_k(t) = \int_0^t B_\alpha(\lambda_k, \eta) f_k(t - \eta) d\eta.$$

The uniqueness of the solution $u(t)$ of problem (3) follows from the uniqueness of the solutions $\omega(t)$ and $w(t)$.

If $\beta \in (0, 1)$ and $K_0$ is not empty, then

$$u(t) = \sum_{k \notin K_0} \left[ \frac{\varphi_k - \omega_k(t_0)}{B_\alpha(\lambda_k, t_0) - \beta} B_\alpha(\lambda_k, t) + \omega_k(t) \right] v_k + \sum_{k \in K_0} h_k B_\alpha(\lambda_k, t) v_k, \tag{37}$$

where $h_k$ are arbitrary numbers. The corresponding orthogonality conditions have the form

$$(\varphi, v_k) = (\omega(t_0), v_k), \ k \in K_0. \tag{38}$$

In particular, if

$$(\varphi, v_k) = 0, \ (f(t), v_k) = 0, \ \text{for all } 0 \le t \le t_0, \ k \in K_0, \tag{39}$$

then the orthogonality conditions (38) are satisfied.

Thus we have proved the following statement.

**Theorem 3.** *Let $\varphi \in H$ and $f(t) \in C([0, T]; D(A^\varepsilon))$ for some $\varepsilon \in (0, 1)$. If $\beta \notin [0, 1)$ or $\beta \in (0, 1)$, but $K_0$ is empty, then problem (3) has a unique solution and this solution has the form (36). If $\beta \in (0, 1)$ and $K_0$ is not empty, then a solution to the problem (3) exists if the orthogonality condition (39) is satisfied. In this case, the solution is not unique and it can be represented as (37) with arbitrary coefficients $h_k$, $k \in K_0$. Moreover, there exist constants $C_\beta > 0$ and $C_\varepsilon > 0$ such that the following coercive estimate holds:*

$$||\partial_t w(t)||^2 + ||Aw(t)||^2 + ||\partial_t^\alpha Aw(t)||^2 \le C_\beta t^{-2(2-\alpha)} ||\varphi||^2 + C_\varepsilon \max_{t \in [0,T]} ||f||_\varepsilon^2, \quad 0 < t \le T.$$

Note that the validity of the assertions in Theorems 2 and 3 requires that the orthogonality conditions (26) and (39) be satisfied, respectively. In light of these conditions a natural question arises: how restrictive are these orthogonality conditions? To answer this question, consider the following example.

Let a bounded domain $\Omega \subset \mathbb{R}^N$ have a sufficiently smooth boundary $\partial\Omega$. Consider the operator $A_0$, defined in $L_2(\Omega)$ with domain of definition $D(A_0) = \{f \in C^2(\Omega) \cap C(\overline{\Omega}) : f(x) = 0, x \in \partial\Omega\}$ and acting as $A_0 f(x) = -\triangle f(x)$. Then, as is known (see, e.g., [32]), $A_0$ has a complete in $L_2(\Omega)$ system of orthonormal eigenfunctions $\{v_k(x)\}$ and a countable set of nonnegative eigenvalues $\lambda_k$ ($\to +\infty$) and $\lambda_1 = \lambda_1(\Omega) > 0$.

Let $A$ be the operator, acting as $Af(x) = \sum \lambda_k f_k v_k(x)$ with the domain $D(A) = \{f \in L_2(\Omega) : \sum \lambda_k^2 f_k^2 < \infty\}$. Then it is not hard to verify that $A$ is a positive self-

adjoint extension in $L_2(\Omega)$ of the operator $A_0$. Therefore, one can apply Theorems 2 and 3 to the following problem:

$$\begin{cases} \partial_t w(x,t) - (1 + \gamma \partial_t^\alpha)\triangle w(x,t) = 0, & x \in \Omega, \quad 0 < t \leq T, \ 0 < \alpha < 1; \\ w(x,t_0) = \beta w(x,0) + \psi(x), \ x \in \Omega, \quad 0 < t_0 \leq T; \\ w(x,t) = 0, \quad x \in \partial\Omega, \quad 0 < t \leq T, \end{cases} \qquad (40)$$

Suppose $\beta \in (0,1)$ and $t_0 \in (0,T]$ satisfies the conditions of Lemma 6. Then, according to Lemma 6, only one eigenvalue can satisfy Equation (17). Let this number be $\lambda_1$, i.e., $B_\alpha(\lambda_1, t_0) = \beta$. We note also that the multiplicity $\lambda_1$ is equal to one.

Therefore, applying Theorem 2, we have that problem (40) has a solution for any function $\psi \in L_2(\Omega)$, if and only if

$$\psi_1 = \int\limits_\Omega \psi(x)v_1(x)dx = 0.$$

In other words, the first Fourier coefficient of $\psi(x)$ must be zero. In this case, the solution of the problem is not unique and all solutions can be represented in the series form

$$w(t) = \sum_{k=2}^\infty \frac{\psi_k}{B_\alpha(\lambda_k, t_0) - \beta}B_\alpha(\lambda_k, t)v_k + hB_\alpha(\lambda_1, t)v_1,$$

which converges in the norm of $L_2(\Omega)$ uniformly in $t \in [0,T]$. Here $h$ is an arbitrary real number.

## 5. Conclusions

In this paper, for the Rayleigh–Stokes equation, we study a new time-nonlocal problem, i.e., in problem (1), instead of the initial condition $u(x,0) = \varphi(x)$, we consider the nonlocal condition $u(x,t_0) = \beta u(x,0) + \varphi(x), 0 < t_0 \leq T$. Moreover, instead of the Laplace operator $(-\Delta)$ (in the Rayleigh–Stokes equation), we consider an arbitrary positive self-adjoint operator $A$. The obtained results are valid for the equation with the Laplace operator under the Dirichlet condition.

The cases of $\beta = 0$ and $\beta = 1$ were studied earlier: if $\beta = 0$, then we obtain a well-known time backward problem that has a unique solution, but the solution is not stable. If $\beta = 1$, then the problem becomes "good", i.e., there is a unique solution and it is stable (see [18]).

The following natural question arises: for what values of $\beta$ is this non-local problem well-posed? This paper provides a comprehensive answer to this question. It turns out that the critical values of the parameter $\beta$ lie on the half-interval $[0,1)$. If $\beta \notin [0,1)$, then the problem is well-posed in the sense of Hadamard: there is a unique solution and it continuously depends on the data of the problem; if $\beta \in (0,1)$ (the case of $\beta = 0$ is considered in [18]), then the well-posedness of the problem depends on the location of the eigenvalues of the Laplace operator. Namely, if the set $K_0$, defined above, is empty, then the problem is again well-posed in the sense of Hadamard. If $K_0$ is not empty, then necessary and sufficient conditions are found guarantying the existence of a solution, but in this case the solution is not unique.

**Author Contributions:** Conceptualization, R.A. and S.U.; methodology, R.A.; validation, R.A., O.M. and S.U.; formal analysis, R.A.; investigation, R.A., O.M. and S.U.; resources, R.A.; writing—original draft preparation, R.A.; writing—review and editing, R.A., O.M. and S.U. All authors have read and agreed to the published version of the manuscript.

**Funding:** This research received no external funding.

**Data Availability Statement:** Not applicable.

**Acknowledgments:** Authors are grateful to Sh. A. Alimov for discussions of the results of the paper. Authors are also thankful to the editor and anonymous reviewers for their valuable comments.

**Conflicts of Interest:** The authors declare no conflict of interest.

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
