# Peer review of "A Non-Local Problem for the Fractional-Order Rayleigh–Stokes Equation"

_fractalfract, doi:10.3390/fractalfract7060490_

Round 1
Reviewer 1 Report
The authors present in this paper a type of initial non-local condition for the fractional model of a generalized second-class fluid flow which be represented as the Rayleigh-Stokes problem with a time-fractional derivative. This initial condition replaces the conventional one described in the paper.
This is a continuation of a previous work and now they intend to investigate the dependence of well-posedness, in the Hadamard sense, for the problem relative to a beta parameter taken between 0 and 1.
The authors made an introduction with a good review of the state of the art. This is an important subject, considering the applications, also described by the authors and it seems to me well treated by them and well grounded in the theoretical aspects.
Although the motivation for the problem under study is well done, I would like to see a conclusion at the end as well as clarification of some aspects such as:
1. For the theorem 2, it is not clear what happens when beta is close to zero for the problem (24).
3. It was also not clear for me the equivalence of solutions between problems (23) +(24) and (3)
2. The authors use theorem 3 as a conclusion, but I think a further conclusion is essential, since the authors have made a good introduction.
There are also minor mistakes like "Direchlet" at line 70
Globally, the matter is well handled and justified by what, in my opinion, deserves to be published after the clarifications mentioned have been made.
Reviewer 2 Report
In the paper “A non-local problem for the fractional order Rayleigh-Stokes equation” by Ravshan Ashurov et. all. the authors the solvability of а nonlocal boundary value problem for the fractional version of Rayleigh-Stokes equation is studied. The authors obtain the conditions under which this solution exists, as well as the conditions under which it is unique
I have a few principled remarks:
1) Estimate (18) is valid only for sufficiently large \lambda. This requirement is stated in lemma 5 formulation, and it is worth noting this after formula (18) for clarification. However with such a condition on \lambda, inequality (22) is not obvious for \lambda_1. The authors need to make this point clear.
2) It is not entirely clear what is meant by the statement in Remark 3: “Under the conditions of this lemma, only one eigenvalue may satisfy equation (20)” . Is it unique for each t_0, or is it unique at all? In the second case, is it possible to specify it?
3) Whence it follows that the denominator (30) does not turn to 0?
There are also typos
Line 179 (17)) - extra parenthesis
Line 241 - there should be a link to the formula (31)
Line 261 space missing after (24)
I think that an article can be accepted for publication in “Fractal and Fractional” only after correcting principled remarks
Round 2
Reviewer 2 Report
I think that after revision the authors of the paper “A non-local problem for the fractional order Rayleigh-Stokes equation” have made corrections according to all the comments and the article can be accepted for publication in “Fractal and Fractional”